# Thoracic Mobilization and Respiratory Muscle Endurance Training Improve Diaphragm Thickness and Respiratory Function in Patients with a History of COVID-19

**DOI:** 10.3390/medicina59050906

**Published:** 2023-05-09

**Authors:** Yang-Jin Lee

**Affiliations:** Department of Physical Therapy, Gyeongbuk College, 77 Daehang-ro, Yeongju-si 36133, Gyeongsangbuk-do, Republic of Korea; ptyangjin2@naver.com; Tel.: +82-54-630-5265

**Keywords:** COVID-19, diaphragm, thoracic mobilization, respiratory function

## Abstract

*Background and Objectives*: Common problems in people with COVID-19 include decreased respiratory strength and function. We investigated the effects of *thoracic mobilization and respiratory muscle endurance training* (TMRT) and lower limb ergometer (LE) training on diaphragm thickness and respiratory function in patients with a history of COVID-19. *Materials and Methods*: In total, 30 patients were randomly divided into a TMRT training group and an LE training group. The TMRT group performed thoracic mobilization and respiratory muscle endurance training for 30 min three times a week for 8 weeks. The LE group performed lower limb ergometer training for 30 min three times a week for 8 weeks. The participants’ diaphragm thickness was measured via rehabilitative ultrasound image (RUSI) and a respiratory function test was conducted using a MicroQuark spirometer. These parameters were measured before the intervention and 8 weeks after the intervention. *Results*: There was a significant difference (*p* < 0.05) between the results obtained before and after training in both groups. Right diaphragm thickness at rest, diaphragm thickness during contraction, and respiratory function were significantly more improved in the TMRT group than in the LE group (*p* < 0.05). *Conclusions*: In this study, we confirmed the effects of TMRT training on diaphragm thickness and respiratory function in patients with a history of COVID-19.

## 1. Introduction

Common COVID-19 infection symptoms include cough, fever (37.5 °C or higher), fatigue, and shortness of breath, while other reported symptoms include weakness, malaria, respiratory distress, muscle pain, and sore throat. As such, the symptoms of COVID-19 patients range from asymptomatic to severe respiratory failure, and about 10% have severe dyspnea and abnormal findings of ground glass shadows in chest computed tomography [1,2,3]. 

In patients with respiratory problems, reduced respiratory efficiency and altered respiratory mechanisms should be corrected by maintaining adequate chest expansion, ventilation, and lung capacity [4].

Joint mobilization exercises of the spinal segments improve muscle efficiency by reducing the excessive use of and strengthening the erector spinae muscle; additionally, these exercises improve performance by allowing the use of appropriate muscles [5]. Moreover, improving the mobility of the muscles surrounding the joint can help optimize joint movement [6]. Watchie (2010) [7] reported that joint mobilization exercises for the thoracic region and backbone resolve ventilation inefficiency caused by pump dysfunction of the chest. Magee (2014) [8] suggested that the correction of chest cage deformations and exercises to improve chest wall flexibility should be performed to relieve the pressure on the lung parenchyma before irreversible damage to the pulmonary blood vessels occurs. Furthermore, Kim and Kim (2022) [9] observed significant differences in respiratory function and diaphragm muscle thickness after thoracic and lumbar stabilization exercises and upper extremity ergometer breathing training in 30 patients who had recovered from COVID-19 compared to the control group. 

Respiratory exercise during hospitalization is a conservative treatment modality and is important for increasing respiratory muscle strength, coughing ability, chest wall mobility, and pulmonary ventilation [10]. Respiratory exercises include breathing using the diaphragm, which is the main inspiratory muscle, exhaling through pursed lips to reduce pain, and evoked spirometry to strengthen the inspiratory muscles [11]. Inspiratory muscle training (IMT) improves muscle strength and endurance by applying a load to the transverse and auxiliary inspiratory muscles [12]. IMT is an effective and safe method for improving respiratory function, cardiorespiratory capacity, activities of daily living, and quality of life; relieving dyspnea; and enhancing endurance in patients with brain injuries [13,14]. 

By comparing exhalation exercise, inhalation exercise, and interventional training that combines inhalation and exhalation, Tout (2013) [15] showed that IMT helps improve pulmonary function. Moodie et al. (2011) [12] reported that applying a load to the diaphragm and synergistic inspiratory muscles helps improve muscle strength and endurance. Nield et al. (2007) [16] reported improved physical function and improved dyspnea in 40 patients with chronic obstructive pulmonary disease after a 12-week pursed-lip breathing intervention, whereas Izadi-avanji and Adib-Hajbaghery (2011) [17] reported an improvement in pulmonary and respiratory function and quality of life.

Spinal joint mobilization and respiratory muscle endurance training are crucial for increasing diaphragm thickness and improving respiratory function in patients with COVID-19. Because no such study has been performed, we examined the effects of spinal joint mobilization and respiratory muscle endurance training on diaphragm thickness and respiratory function in patients previously diagnosed with COVID-19.

## 2. Materials and Methods

### 2.1. Participants

Among subjects who had been diagnosed with COVID-19 for one month, this study was conducted on subjects who met the selection criteria. The study included 30 volunteers who were recruited through an advertisement at Hospital B, a general hospital located in Gyeonggi-do. The inclusion criteria were as follows: (1) history of COVID-19 at least 1 month prior; (2) forced vital capacity (FVC) <80% of the predicted normal value and not receiving specific treatment; (3) no cardiovascular disease or depression; (4) a score of at least 24 points on the Mini-Mental State Examination—Korean (MMSE-K) and the ability to communicate and follow instructions; and (5) the provision of voluntary written consent before study participation. The exclusion criteria were as follows: (1) congenital or acquired thoracic cage deformities; (2) a history of undergoing chest or abdominal surgery; (3) the inability to perform respiratory mechanisms; and (4) orthopedic diseases of the trunk. 

The study was conducted from November to December 2021 and included 30 volunteers, who were divided into thoracic mobilization and respiratory muscle endurance training (TMRT, *n* = 15) and lower limb ergometer (LE, *n* = 15) groups according to the experimental objective. To minimize selection bias, the groups were divided based on random assignment using a computer. The training program was conducted for 30 min 3 times a week for 8 weeks. Assessments were conducted before and eight weeks after the experiment. Pre- and post-test, rehabilitative imaging ultrasound was used to measure changes in diaphragm thickness. In addition, a diagnostic spirometer was used to measure respiratory function in terms of FVC, forced expiratory volume in 1 s (FEV1), and maximum expiratory rate. The study adhered to the Helsinki Declaration principles and was approved by the Gimcheon University Institutional Review Board (No: GU-202104-HRa-05-02; 21 June 2021).

The sample size determination was founded on data collected from a pilot study. We calculated the sample size using G*Power 3.1.9.7 software (Heinrich-Heine-University Düsseldorf, version 3.1.9.7, Düsseldorf, Germany). The effect size variable was right-side diaphragm contraction. The input parameters were group 1 (mean: 0.03, SD: 0.01) and group 2 (mean: 0.02, SD: 0.01). Thus, a total of 30 study subjects were calculated, (15 in each group); the effect size D was 1.2649111, the alpha error was 0.05, and the power was 0.90.

### 2.2. Intervention

#### 2.2.1. TMRT Group

Spine mobilization and respiratory muscle endurance training were conducted in this group. 

Spine mobilization was conducted for 15 min using the method proposed by Maitland (2005) [18]. Joint mobilization was applied according to the level of pain and restriction of movement: Grade I—low-amplitude vibration at the beginning of the range of motion; Grade II—high-amplitude vibration at the midpoint of the range of motion; Grade III—high-amplitude vibration at the end of the range of motion; and Grade IV—low-amplitude vibration at the end of the range of motion. The volunteers were asked to lie comfortably in the prone position while the upper part of the table was lowered to allow slight bending of the spine. While standing next to the volunteer, the therapist placed their metacarpals, lateral, or tibia on the spinal processes of the volunteer’s thoracic vertebrae. Subsequently, Grade II–III joint mobilization was applied by extending the arm straight so that the shoulders were directly above the spine and delivering a load through the arm to the hand [18]. Central and unilateral posteroanterior and transverse mobilization were applied to the spinal segments with reduced mobility (Figure 1).

Respiratory muscle endurance training was conducted for 15 min. A K5 device (POWER Breathe^®^, Southam, UK) was used to measure the maximum inspiratory pressure while simultaneously performing the respiratory exercises. The volunteers were asked to sit with their backs straight and to inhale quickly through the mouthpiece after forcefully exhaling all the residual air from the lungs. This was repeated 30 times (one set) for three sets with a 1 min rest between sets. If the individuals complained of dizziness or fatigue, the session was resumed after a short break (Figure 2).

#### 2.2.2. LE Group

The LE group performed ergometer exercises of the lower limbs. A New 3000 device (Shin Gwang, Paju, South Korea) was used for 30 min aerobic exercises. Exercise intensity was 40–50% of the maximum heart rate (HRmax) through weeks 1–4, 50–55% of the HRmax through weeks 5–8, and 55–60% of the HRmax through weeks 9–12. All individuals were asked to wear a heart rate monitor to maintain exercise intensity [19].

A pre-test was conducted before the first intervention, and a post-test was conducted after all interventions were completed. An assistant stood by for safety reasons at all times in case of a fall. A mat was prepared for the rest intervals. The devices were disinfected to prevent infection once the measurements were completed.

### 2.3. Evaluation

#### 2.3.1. Diaphragm Thickness 

A Rehabilitative Ultrasound Image (RUSI) digital image analyzer was used to measure diaphragm thickness. A MYSONO U5 (Samsung Medicine, Seoul, South Korea) real-time ultrasound imaging device was used for image collection. All tests used a 7.5 MHz linear transducer, 6–8.5 MHz frequency modulation, and a 20–80-grain range. The individuals were first asked to comfortably lie down, and the space between the 8th and 9th intercostal muscles along the right axillary line was marked. Subsequently, while in the supine position, the transducer was moved perpendicular to the chest wall to measure the space between the 8th and 9th intercostal muscles in a two-dimensional coronal plane. The individuals were asked to repeat the maximum inhalation and exhalation process three times to accurately measure the diaphragm thickness at maximum exhalation (at rest) and maximum inhalation (at contraction). The changes in thickness were measured, and the mean of the three measurements was calculated. 

#### 2.3.2. Respiratory Function 

A PC-based spirometer (MicroQuark, Cosmed, Italy) was used to measure pulmonary function. First, the volunteers were asked to sit comfortably on a bed and hold a personal mouthpiece between their teeth with their lips covering it to prevent exhalation through the nose. The measurement variables included FVC, FEV1, and peak expiratory flow rate (PEF) after measuring maximal expiratory effort using a spirogram. The mouthpiece was immediately separated and disinfected with alcohol for hygiene purposes after each measurement. 

### 2.4. Data Analysis

PASW for Windows (version 20.0; IBM-SPSS, Seoul, Republic of Korea) was used for all statistical analyses. The general characteristics and dependent variables were compared between the two groups before training using the chi-squared (gender) and independent t-tests (age, height, weight, and BMI). The Shapiro–Wilk test was used to assess normality. Independent t-tests were used to compare the differences in changes between the two groups after 8 weeks of training, while paired t-tests were used to examine the differences in training between the two groups following the intervention period. The significance level for all statistical tests was α = 0.05.

## 3. Results

### 3.1. General Characteristics of the Research Subjects

The general characteristics of the subjects in the TMRT and LE groups were homogeneous (Table 1). 

### 3.2. Changes in Diaphragm Thickness

The left and right diaphragm thicknesses at rest differed significantly before and after the experiment in the two groups. The change in right diaphragm thickness at rest before and after the test differed significantly in the TMRT group (change value: 0.01 cm) compared to the LE group (change value: 0.01 cm). However, the change in left diaphragm thickness at rest before and after the test did not differ significantly between intervention methods (Table 2).

The left and right diaphragm thickness during contraction differed significantly before and after the experiment in both groups. The change in left and right diaphragm thickness differed significantly in the TMRT group (change values: 0.04 cm and 0.05 cm) compared to the LE group (change values: 0.03 cm and 0.03 cm) (Table 2). The left and right diaphragm thicknesses at rest differed significantly before and after the experiment in the two groups. However, the change in left and right diaphragm thickness at rest before and after the test did not differ significantly between intervention methods (Table 2).

The left and right diaphragm thickness during contraction differed significantly before and after the experiment in both groups. The change in left and right diaphragm thickness differed significantly in the TMRT group (change values: 0.04 cm and 0.05 cm) compared to the LE group (change values: 0.03 cm and 0.03 cm) (Table 2).

### 3.3. Change in Respiratory Function

FVC, FEV1, and PEF differed significantly in the two groups before and after the experiment (Table 3). Regarding the change in FVC, the TMRT group (change value: 0.25 L) showed a significant difference compared to the LE group (change value: 0.09 L). Regarding the change in FEV1, the TMRT group (change value: 0.34 L) showed a significant difference compared to the LE group (change value: 0.16 L). Finally, the TMRT group (change value: 0.31 L) showed a significantly different change in PEF compared to the LE group (change value: 0.17 L) (Table 3).

## 4. Discussion

This study divided patients who experienced COVID-19 into TMRT (15 volunteers) and LE (15 volunteers) groups to examine their diaphragm thickness and respiratory function. The results showed significantly greater differences in the TMRT group than in the LE group. 

Long-term sequelae of COVID-19 (also known as post-COVID condition, long-term COVID, long COVID, and chronic COVID) are defined as the persistence of symptoms and signs for at least 12 weeks after COVID-19 that are not explained by other diagnoses. However, there remains no global consensus on the definition; furthermore, newly emerging late-onset sequelae and changes in symptoms or conditions are also referred to as long-term sequelae of COVID-19 [20]. Among these sequelae, dyspnea reportedly occurs in one-quarter of patients following COVID-19 [21]. During a 2-month observation period, persistent dyspnea occurred in about half of patients following COVID-19, with one-third showing persistent cough and only 27% showing improvement in chest radiographs [22]. These results suggest that patients with COVID-19 require interventions for respiratory function improvement.

Thoracic mobilization increases the following: facet joint sliding of the thoracic vertebrae, thoracic flexibility by inducing chest expansion by normalizing the joint capsule, thoracic movement during inhalation, and thoracic expansion. Additionally, thoracic mobilization also helps improve lung function [23,24]. Therefore, herein, we examined the effects of TMRT and LE on diaphragm thickness and respiratory function in patients who had been diagnosed with COVID-19.

The results of this study showed significant changes in pre- and post-intervention diaphragm thickness in the TMRT and LE groups; however, no statistically significant differences were identified between the groups. This finding was consistent with that reported by Kim et al. (2013) [19] on improved diaphragm thickness in patients with stroke following breathing retraining. Kaneko et al. (2010) [25] reported that changes in diaphragm thickness were closely related to pulmonary capacity during maximum inhalation. Thus, the increased diaphragm thickness in the present study may have improved exercise performance by positively affecting the inspiratory muscles involved in physical performance [26]. The change in diaphragm thickness observed in this study may have had a positive effect through the increase in thoracic vertebrae mobility and diaphragm contraction alteration; the respiratory muscle endurance training provided active respiratory exercises. 

Forced vital capacity (FVC), forced expiratory volume in one second (FEV1), and peak expiratory flow (PEF) are used as indicators to estimate levels of respiratory function [27]. The results of the present study showed significant changes in respiratory function in the TMRT and LE groups, as well as significant differences between the groups. These findings are consistent with those of a prospective cohort study by Gloeckl et al. [28], which reported improvements in 6 min walking distance, FVT, and FEV1 following respiratory rehabilitation in patients with severe COVID-19. Our study findings are also consistent with those of Liu et al. (2020) [29], who reported significant improvements in respiratory function indicators, such as FVC and FEV, and 6 min walking distance in the intervention group (where a threshold-resistant expiratory muscle strengthening device was incorporated with coughing exercises, stretching, and diaphragm training in older adults with COVID-19) compared to the control group. In addition, Mueller et al. [30] showed similar findings to a study in which the ratio of forced expiratory volume for 1 s-to-forced vital capacity and forced expiratory volume for 1 s significantly increased after a breathing exercise in spinal cord injury patients. The thoracic vertebrae mobilization exercises in this study induced thoracic vertebrae and rib cage movement, while the respiratory muscle endurance training strengthened the diaphragm, increasing the inflow and outflow of air, thereby resulting in respiratory function changes. 

The limitations of this study include the following: (1) the effects of participant-dependent variables could not be completely ruled out due to the environmental factors in their daily lives; (2) the interpretation and generalization of the study results for diaphragm thickness and respiratory function changes in patients with COVID-19 are limited, as these individuals were selected based on inclusion and exclusion criteria; (3) there was no control group, so there was a lack of access to data on improvement in respiratory function over time; and (4) we did not consider the possibility that the small sample and BMI values might have confounded the results. Additional research is needed to evaluate the mobilization of various parts involved in respiratory function, different methods of respiratory muscle endurance training, and various dependent variable assessment tools.

## 5. Conclusions

This study’s findings indicate that TMRT can be considered a potential method to improve diaphragm thickness and respiratory function in patients with COVID-19. Diversified TMRT will need to be developed for broader application of the combined approach as a therapeutic intervention for the functional recovery of patients with long-term COVID-19.

## Figures and Tables

**Figure 1 medicina-59-00906-f001:**
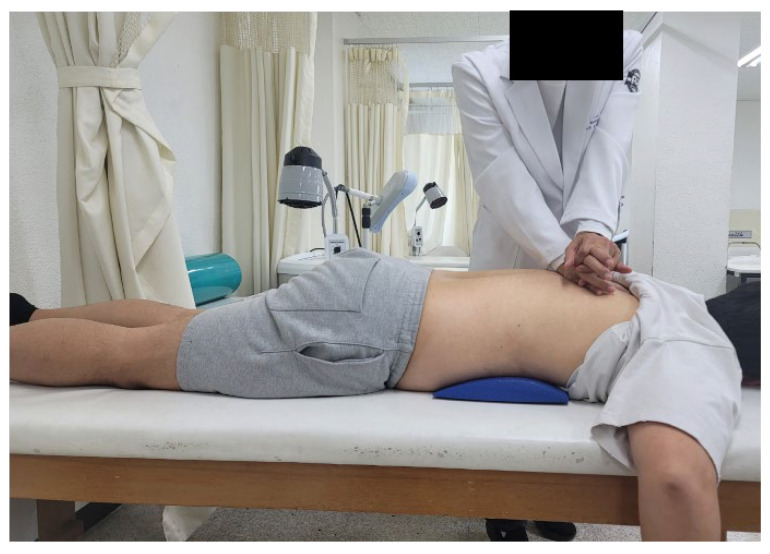
Thoracic spine mobilization exercise.

**Figure 2 medicina-59-00906-f002:**
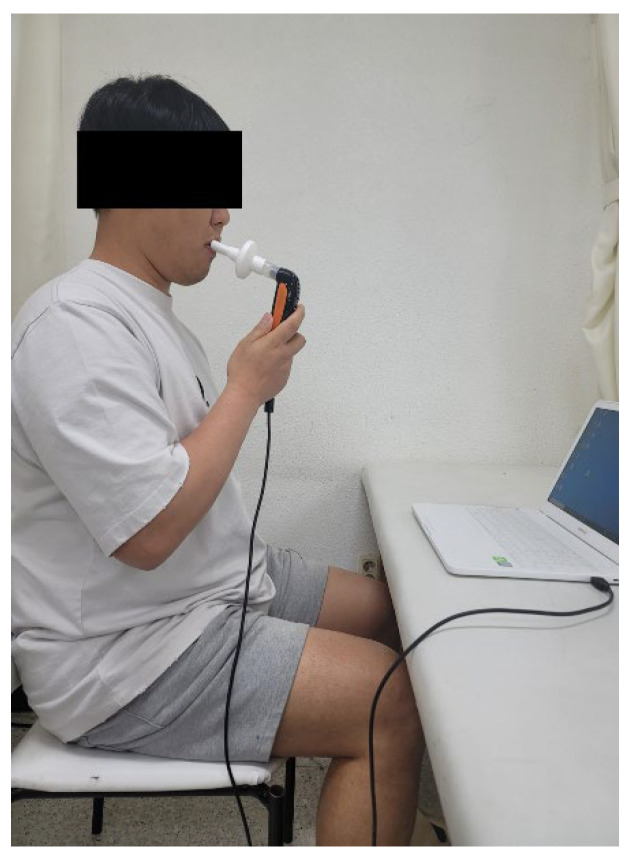
Respiratory muscle endurance training.

**Table 1 medicina-59-00906-t001:** Demographic characteristics of patients (*n* = 30).

	TMRT Group (*n* = 15)	LE Group (*n* = 15)	*p*-Value
Age (years)	23.13 ± 1.06	22.33 ± 1.45	0.12
Height (cm)	166.48 ± 7.99	167.73 ± 8.72	0.68
Weight (kg)	68.10 ± 14.55	69.40 ± 14.08	0.81
BMI	6.50 ± 1.43	7.10 ± 0.99	0.93
Gender (male/female)	9 (60.0%)/6 (40.0%)	8 (53.3%)/7 (46.7%)	0.71

Values are presented as means ± standard deviation. TMRT group—thoracic mobilization and respiratory muscle endurance training group; LE group—lower limb ergometer group.

**Table 2 medicina-59-00906-t002:** Comparison of diaphragm thickness of TMRT and control groups.

Measures	TMRT Group (*n* = 15)	LE Group (*n* = 15)	*t*	*p*	
Left side diaphragm rest (cm)	Pre-test	0.20 ± 0.05	0.19 ± 0.04	0.36	0.72
	Post-test	0.21 ± 0.04 ^a^	0.20 ± 0.04 ^a^		
	Change	0.01 ± 0.01	0.01 ± 0.01	2.033	0.05
Right side diaphragm rest (cm)	Pre-test	0.19 ± 0.05	0.19 ± 0.04	0.17	0.87
	Post-test	0.21 ± 0.05 ^a^	0.20 ± 0.04 ^a^		
	Change	0.01 ± 0.00	0.01 ± 0.00	1.06	0.03 *
Left side diaphragm contraction (cm)	Pre-test	0.59 ± 0.04	0.58 ± 0.03	0.42	0.68
	Post-test	0.63 ± 0.04 ^a^	0.61 ± 0.03 ^a^		
	Change	0.04 ± 0.01	0.03 ± 0.00	4.68	0.00 *
Right side diaphragm contraction (cm)	Pre-test	0.59 ± 0.03	0.59 ± 0.03	0.17	0.86
	Post-test	0.64 ± 0.04 ^a^	0.61 ± 0.03 ^a^		
	Change	0.05 ± 0.02	0.03 ± 0.04	5.78	0.00 *

Values are presented as means ± standard deviation; * *p* < 0.05; ^a^ Significant differences between pre- and post-test (*p* < 0.05). TMRT group—thoracic mobilization and respiratory muscle endurance training group; LE group—lower limb ergometer group.

**Table 3 medicina-59-00906-t003:** Comparison of respiratory function of TMRT and control groups.

Measures		TMRT Group (*n* = 15)	LE Group (*n* = 15)	*t*	*p*
Force vital capacity (L)	Pre-test	4.10 ± 0.44	4.09 ± 0.44	0.06	0.95
	Post-test	4.35 ± 0.36 ^a^	4.18 ± 0.42 ^a^		
	Change	0.25 ± 0.19	0.09 ± 0.08	3.15	0.00 *
Forced expiratory volume in the one second (L)	Pre-test	4.00 ± 0.44	3.96 ± 0.22	0.30	0.77
	Post-test	4.34 ± 0.31 ^a^	4.13 ± 0.26 ^a^		
	Change	0.34 ± 0.19	0.16 ± 0.17	2.72	0.00 *
Peak expiratory flow (L)	Pre-test	4.71 ± 0.90	4.62 ± 0.58	0.34	0.74
	Post-test	5.02 ± 0.76 ^a^	4.78 ± 0.47 ^a^		
	Change	0.31 ± 0.20	0.17 ± 0.15	2.20	0.00 *

Values are presented as means ± standard deviation; * *p* < 0.05; ^a^ Significant differences between pre- and post-test (*p* < 0.05). TMRT group—thoracic mobilization and respiratory muscle endurance training group; LE group—lower limb ergometer group.

## Data Availability

Data is unavailable due to privacy or ethical restrictions.

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
