# Peer review of "Thoracic Mobilization and Respiratory Muscle Endurance Training Improve Diaphragm Thickness and Respiratory Function in Patients with a History of COVID-19"

_medicina, 2023, doi:10.3390/medicina59050906_

Round 1

Reviewer 1 Report

1- Tables are recommended to be summarized.

2- some figures could be provided to demonstrate the mechanisms.

3- Discussions should be strengthen by referring to other studies.

Author Response

1- Tables are recommended to be summarized.

: anser) table Edited.

2- some figures could be provided to demonstrate the mechanisms.

: anser) Added to the text.
(Figure 1. Thoracic spine mobilization exercise., Figure 2. Respiratory muscle endurance training.)

3- Discussions should be strengthen by referring to other studies.

: anser) Added content.

Forced vital capacity (FVC), forced expiratory volume in the one second (FEV1), and peak expiratory flow (PEF) have been used as indicators to estimate the level of respiratory function (Kisner & Colby, 2017)

In addition, Mueller et al. (2008) showed similar findings to a study in which the ratio of forced expiratory volume for 1 second to forced vital capacity and forced expiratory volume for 1 second significantly increased after breathing exercise in spinal cord injured patients.

Author Response

In the study: Effects of thoracic mobilization and respiratory muscle endurance training on diaphragm thickness and respiratory function in patients with a history of COVID-19, the authors investigated the effects of TR and LE training on the thickness of diaphragm and lung function in subjects who had COVID19. Totally 30 subjects performed thoracic mobilization and respiratory muscle endurance training or lower limb ergometer training, 15 subjects in each group. The authors revealed significant difference in the thickness of diaphragm and lung function in both groups before and after the trainings, but the difference of improvement was

significant only in TR group. The study is interesting and can have value in improvement of respiratory function in subjects who had COVID 19.

Minor revision:

  1. I would change the title to:

Thoracic mobilization and respiratory muscle endurance training improves diaphragm thickness and respiratory function in patients with a history of COVID-19.

: anser) Edited.

Thoracic mobilization and respiratory muscle endurance training improves diaphragm thickness and respiratory function in patients with a history of COVID-19.

  1. There are incorrections in the abstract, some sentences are not understandable.

Suggested changes in the abstract

: anser) Edited.

Abstract: We investigated the effects of thoracic mobilization and respiratory muscle endurance (TR) training, and lower limb ergometer (LE) training on diaphragm thickness and respiratory function in patients with a history of COVID-19. Totally, 30 patients were randomly divided into TR training group and LE training group. TR group performed training of Thoracic mobilization and Respiratory muscle endurance for 30 min 3 times a week for 8 weeks. LE group performed training of Lower limb ergometer for 30 min 3 times a week for 8 weeks. The diaphragm thickness was measured by RUSI and the respiratory function test was measured by spirometer MicroQuark. There was a significant difference (P < 0.05) between before and after training in both groups. However, TR group showed significant improvement in both, diaphragm thickness and respiratory function. In this study, we confirmed effects of TR training on the diaphragm thickness and respiratory function in patients with a history of COVID-19.

  1. Introduction, the first and second paragraph, the authors should avoid repeating

: anser) Edited.

Common COVID-19 infection symptoms include cough, fever (37.5°C or higher), fatigue, and shortness of breath, while other reported symptoms include weakness, malaria, respiratory distress, muscle pain, and sore throat. As such, the symptoms of COVID-19 patients ranged from asymptomatic to severe respiratory failure, and about 10% had severe dyspnea and abnormal findings of ground glass shadows in chest computed tomography

  1. line 161 tests were used to….

: anser) Edited.

  1. line 176, 177, within brackets, at the end add respectively, to relate the numbers to left and right diaphragm thickness.

: anser) Edited.

  1. Discussion, the first paragraph, why there is 20 subjects per group, it is previously stated that there is 15 subjects per group

: anser) Edited.

Reviewer 3 Report

Design: To test the hypothesis, I think you would also need a control group with patients not receiving any specific training at all (i.e., respiratory function may be improved- over time - due to the recovery from COVID). 

Characteristics: Could you provide more information about the time since the COVID infection and the enrollment in the study? 

Discussion: In the discussion, it is stated that two groups of 20 patients were formed, but the methods state 2x15. Generalizability is low given the small sample and the characteristics of the population in this study (relatively young adults). Did you consider the confounding effect of BMI (>25) on the results? 

Author Response

  1. Design: To test the hypothesis, I think you would also need a control group with patients not receiving any specific training at all (i.e., respiratory function may be improved- over time - due to the recovery from COVID). 

: ã€€anser) The design details for the control group were described as limitations.

(There was no control group, so there was a lack of access to improvement in respiratory functionover time.)

  1. Characteristics: Could you provide more information about the time since the COVID infection and the enrollment in the study? 

: ã€€anser) : Content has been inserted.

Among the subjects who had been diagnosed with COVID-19 for one month, the study was conducted through subjects who met the selection criteria.

  1. Discussion: In the discussion, it is stated that two groups of 20 patients were formed, but the methods state 2x15. Generalizability is low given the small sample and the characteristics of the population in this study (relatively young adults). Did you consider the confounding effect of BMI (>25) on the results? 

: ã€€anser)  Content has been modified.

15 patients

: ã€€anser) The contents were inserted at the limitations of the study.

(   4) we did not consider the possibility that the small sample and BMI values might have confounded the results.)

Round 2

Reviewer 1 Report

Lack of novelty and advanced methodology makes the manuscript unsuitable for publication.

Author Response

A method for improving respiratory function in patients experiencing COVID-19 was studied through an approach combining thoracic spine mobility exercises, not just simple breathing exercises. We studied the relationship between these exercises to the activation of the diaphragm and respiration, and we will supplement the lacking approach in the future. thank you.

English proofread file is attached.

Reviewer 3 Report

The results of this study showed significant changes in the pre- and post-intervention

diaphragm thickness in the TMRT and LE groups but; however, no statistically significant

differencedifferences were identified between the groups.

Is this correct following the study findings? 
